# Comparative Impact of Organic Grass-Fed and Conventional Cattle-Feeding Systems on Beef and Human Postprandial Metabolomics—A Randomized Clinical Trial

**DOI:** 10.3390/metabo14100533

**Published:** 2024-10-03

**Authors:** Meghan Spears, Gwendolyn Cooper, Brett Sather, Marguerite Bailey, Jane A. Boles, Brian Bothner, Mary P. Miles

**Affiliations:** 1Department of Food Systems, Nutrition, and Kinesiology, Montana State University, Bozeman, MT 59717, USA; meghanspears@montana.edu; 2Department of Chemistry and Biochemistry, Montana State University, Bozeman, MT 59717, USA; gwendolyn.cooper@student.montana.edu (G.C.);; 3Department of Animal and Range Sciences, Montana State University, Bozeman, MT 59717, USA

**Keywords:** beef, untargeted, metabolomics, cattle-feeding systems, postprandial, pathway enrichment, health implications

## Abstract

Background/Objectives: Cattle-feeding systems may have health implications for consumers of beef products. Organic grass-fed (GRA) and conventional (CON) cattle-feeding systems may result in beef products with differing metabolite profiles and therefore could impact the postprandial metabolomic response of consumers. This study aims to measure whole beef metabolomics and postprandial metabolomic response of consumers between GRA and CON beef to elucidate potential health implications. Methods: This study followed a randomized double-blind crossover design with healthy male and female subjects (*n* = 10). Plasma samples were taken at fasting (0) and postprandially for four hours after consumption of a steak from each condition. Untargeted metabolomic analysis of whole beef and human plasma samples used LC/MS. Multivariate and pathway enrichment analysis in MetaboAnalyst was used to investigate metabolite and biochemical pathways that distinguished CON and GRA. Results: Cattle-feeding systems impacted both postprandial and whole beef steak metabolomic profiles. Metabolites that contributed to this variation included carnitine species (Proionylcarnitine), fatty acids, amino acids (L-valine), and Calamendiol. These metabolites have been associated with oxidative stress, inflammation, and cardiovascular health. Functional pathway enrichment analysis revealed numerous amino acid degradation pathways, especially branched-chain amino acids, and fatty acid degradation that changed throughout the postprandial time course. Conclusions: These findings suggest that CON and GRA cattle-feeding systems differentially impact whole beef metabolomics, as well as consumer postprandial metabolic responses and the associated health implications.

## 1. Introduction

Beef is a commonly consumed meat product in developed nations, making up 21% of global meat product intake in 2022, and has potentially large impacts on human health [1,2]. It serves as an important dietary protein source that contains essential nutrients, including fatty acids and numerous vitamins and minerals [3]. However, the nutrient composition of beef can be altered by the methods used to feed cattle prior to slaughter [4]. Attention has been brought to how the method of cattle-feeding may impact the health outcomes associated with consuming beef. In particular, organic grass-fed (GRA) has been compared to conventional (CON) cattle-feeding systems [5]. One of the main differences between GRA and CON beef is that GRA cattle experience a longer period of time with access to forage/grass in their lifespan [6]. This difference in the diet has been shown to influence the protein, nutrient, and fatty acid composition of the resulting meat product [7]. These nutritional differences and coinciding health impacts make cattle-feeding systems a point of interest when considering dietary choices.

Some of these differences and how they can impact health are known, but there are additional differences between GRA and CON that may induce metabolic changes not previously characterized. For instance, GRA beef has been shown to contain fewer monounsaturated fatty acids (MUFA), as well as more n-3 fatty acids compared to CON beef [4,5]. MUFAs have been associated with healthier glycemic control while n-3 fatty acids can combat inflammation [8,9]. GRA and CON cattle-feeding systems have also been shown to impact the muscle tissue metabolism of the cattle. CON-fed cattle appear to adapt to their abundant energy sources and produce larger fat deposits while GRA cattle adapt to a lower intake of energy by maintaining greater amounts of oxidative enzymes such as succinate dehydrogenase [10]. This was also confirmed via metabolomic analysis demonstrating significant variation in metabolite profiles between pasture and feedlot cattle muscle tissue, specifically greater amounts of ATP and fumarate in pasture-raised cattle [11]. Cattle-feeding systems also have been shown to impact muscle tissue protein content. Significant differences in total protein content have been observed between GRA and CON beef [12]. It has also been shown that CON beef has greater amounts of the essential amino acids leucine and methionine than GRA beef [11]. Essential amino acids are required in the diet and play a vital role in anabolic reactions, especially in aging adults [13]. Lastly, micronutrients like B vitamins, riboflavin, and thiamine have been found in greater amounts in GRA beef [14]. B vitamins are important mediators in cellular mechanisms related to cognitive function and microbiome health [15,16]. This variety of research describes the impact of cattle-feeding systems of whole beef samples and the possible related health implications; however, the human postprandial metabolomic responses to consuming GRA and CON beef are underdeveloped [11].

Postprandial metabolic events induced by the consumption of beef are important to understand because the postprandial state is closely related to overall health [17]. Beef intake, in comparison to other dietary protein sources (plant-based and fish), has been shown to increase postprandial branched-chain amino acid (BCAA) levels and increase satiety, as well as increase 2-aminoadipic acid [18,19]. Postprandial BCAA and 2-aminoadipic acid levels have been associated with increased insulin secretion, making red meat intake an important consideration for consumers with glycemic control concerns [20,21]. Satiety induced by dietary protein sources is also an indicator of body weight maintenance, benefiting metabolic health [22]. This demonstrates the importance of postprandial measurements of beef intake-induced metabolite changes.

There is a dynamic transition from a fasted state to a fed state that is difficult to capture; however, metabolomics has become an accurate and sensitive form of analysis to assess these small and large changes that can occur [23,24]. The use of metabolomics in human nutrition research provides important insight into the connections between the foods we eat and how they contribute to changes in diet-associated biomarkers during a postprandial period [25]. This does not come without challenges, as the metabolomic profile of individuals not only contains diet-related metabolites, but non-nutrient metabolites as well [23,26]. Given how vital it is to maintain a phenotype that is dynamic and flexible, it is imperative to characterize and identify biomarkers reflective of diet [27].

Metabolites, the substrates and products of metabolism provide key insight into essential cellular functions such as signal transduction, energy production, and energy storage [28]. Untargeted metabolomics is a strong analytical method to detect a wide variety of metabolites. Therefore, changes in metabolite abundance or within specific conditions of metabolites provide a powerful tool for identifying and tracking biomarkers reflective of complex physiological responses [28,29]. Untargeted metabolomic analyses aim to characterize all endogenous metabolites in a given matrix with an overall goal of capturing global changes in physiological status, biochemical pathways, or sites of perturbation [30]. Thus, metabolomic analyses provide key information on the metabolic transition from fasting to a fed state. Additionally, the inherent sensitivity of metabolomics enables subtle changes or alterations in biochemical pathways, specifically those changing in response to a nutritional challenge, to be detected [31].

This study aimed to characterize metabolic responses to ingestion of beef and measure metabolic similarities and differences associated with CON- and GRA-fed beef—both within the steaks themselves as well as in the response of human consumers. It was hypothesized that cattle-feeding systems would induce unique metabolomic changes in the composition of beef from these production systems that are reflected in the postprandial metabolic responses as well.

## 2. Materials and Methods

### 2.1. Study Population

Participants were recruited in Bozeman, MT from August 2022 to March 2023 by word of mouth on a rolling basis. Inclusion criteria included being between 18 and 45 years old with a body mass index (BMI) of 18–27 kg/m^2^. Anthropometrics were assessed with segmental multifrequency bioelectrical impedance analysis (Seca mBCA 515, Hamburg, Germany). Participants were excluded if they had preexisting conditions, including allergy to red meat, diabetes, gall bladder condition, or taking medication to lower cholesterol, inflammation, and blood pressure. Screening of potential participants was conducted using REDCap (version 13.10.6) [32].

### 2.2. Experimental Design

The study was a double-blind crossover randomized clinical trial with two conditions, GRA and CON beef. Participants came to the Nutrition Research Laboratory on three occasions for testing. The first visit consisted of an informed consent review followed by a general health questionnaire and anthropometric measurements. The study protocol was approved by the Institutional Review Board at Montana State University (2022-138-EXPIDITED). The informed consent was reviewed verbally in person and informed consent was obtained from all participants prior to their participation. This study was prospectively registered with ClinicalTrial.gov (NCT05460754). The second and third visits occurred at least 7 days apart and included the dietary intervention of steak with fasting and half-hourly blood collection for 4 h postprandially for blood marker analysis. See Appendix A for a summary of the study design. Plasma samples taken at each timepoint were analyzed using LC/MS untargeted metabolomics to assess metabolomic profile differences between CON and GRA. Samples of whole beef from CON and GRA steaks were also collected and analyzed with LC/MC untargeted metabolomics to metabolomic differences between beef from both conditions.

### 2.3. Dietary Intervention and Blood Sample Collection

The order in which participants received steaks at visits 2 and 3 was determined by the randomization protocol. Participants were randomly assigned the order in which steaks from GRA and CON were administered by random number assignment and balanced based on visit number. High and low numbering of each participant and the treatment received was assigned, then sorted three times to determine the order of steak administration. Steaks were blinded with a three-digit code written on the vacuum-sealed package by a member of the research team not involved in data collection or analysis. Researchers, volunteers, and clinical staff aiding in data collection were blinded from condition assignments at visits 2 and 3.

Participants were instructed not to consume any red meat beginning 72 h before visit 2 through to the completion of visit 3, and to avoid fatty fish the day before visit 2 and 3. The diet the day before visit 2 was then replicated by participants before visit 3 to improve comparability.

Prior to visits 2 and 3 participants fasted for 12 h and avoided strenuous activity or the consumption of alcohol for 24 h. An intravenous catheter was inserted in the antecubital vein upon arrival by a phlebotomist or medical doctor. Blood samples were collected into plasma-separating vacutainer tubes containing heparin (BD Vacutainer, Franklin Lakes, NJ, USA) after a 3 mL waste withdrawal followed by a sterile saline flush. The fasting blood sample was taken 15 min after catheter insertion. Postprandial blood samples were collected every 30 min for 4 h beginning 30 min after the participant began eating the study steak. Ad libitum black tea was offered to participants to accompany the steak. Blood markers glucose (GLU) and triglycerides (TG) were measured at fasting and hourly timepoints using whole blood collected in sodium heparin tubes on a Picollo Xpress Chemistry Analyzer lipid panel (Abaxis, Union City, CA, USA). Samples were allowed to clot for 15 min prior to centrifugation (14,515× *g*, 15 min) to separate plasma. The plasma was then aliquoted and frozen at −80 °C for future analysis.

### 2.4. Steak Preparation

Organic, GRA beef strip loin was sourced from B Bar Ranch Big Timber, MT, and CON beef choice strip loin was sourced from a local purveyor. All steaks were cut from the longissimus lumborum muscle and matched for marbling. Two cows per condition were used to source strip loin cuts. Steaks were vacuum sealed and stored at −20 °C until they were needed for participants. All steaks were removed from the freezer and thawed in a refrigerator between 24 and 48 h before the planned time of consumption. An uncooked sample of beef weighing between 2 and 5 g was collected for future metabolic analysis. The pre-cooking weight of the raw steak was recorded. Steaks were cooked on a clamshell grill with a thermometer (ThermoWorks THS-313-158 probe with Therma Waterproof THS-232-101, American Fork, UT, USA) inserted into the thickest part of the meat until the thermometer read 70 °C. A cooked sample of beef weighing 2–5 g was collected for future metabolic analysis. Weight was recorded immediately after taking off the grill. The steak was then wrapped in foil and rested for 5 min. After 5 min of rest, the cooked steak was cut to 170.010 g and served to participants.

### 2.5. Whole Beef Steak Metabolite Extraction

Samples of beef were taken from each cow used to harvest meat for this study (2 for GRA beef, 2 for CON beef). Samples were taken before and after cooking and frozen immediately to halt any potential metabolic changes. For whole beef analysis, 0.1 g of beef was suspended in 1 mL of cold 3:1 MeOH/water with stainless steel beads. Homogenization was achieved using a SPEX SamplePrep tissue homogenizer (SPEX SamplePrep, Metuchen, NJ, USA). Homogenate was removed, placed into a fresh tube, and centrifuged at 16,100× *g* for 10 min at 4 °C to remove cellular debris. To precipitate proteins, the supernatant was removed and 4× the volume of cold acetone was added, vortexed briefly, and then stored at −80 °C overnight. The following day, samples were centrifuged at 16,100× g for 10 min at 4 °C. The supernatant containing the remaining metabolites was transferred to a fresh vial and dried down via vacuum centrifugation. All samples were then resuspended in 1:1 acetonitrile (ACN)/water and immediately analyzed via mass spectrometry.

### 2.6. Plasma Metabolite Extraction

To extract plasma metabolites and precipitate proteins, 100 µL of plasma was added to 400 µL of cold acetone. Samples were then placed at −80 °C overnight. Proteins and any other remaining macromolecules were then pelleted via centrifugation at 16,100× *g* for 15 min at 4 °C. Supernatants containing metabolites were then collected and dried down via vacuum concentration. All samples were stored at −80 °C until subsequent mass spectrometry analysis. All solvents used in metabolite extractions were high-performance liquid chromatography (HPLC) grade or higher.

### 2.7. Untargeted Metabolomic Analysis

Following metabolite extraction of whole beef and human plasma, all samples were analyzed using an Acquity ultra-high-performance liquid chromatography (UPLC) coupled to a Waters Synapt XS (Waters, Milford, MA, USA) operated in positive mode. Metabolites were separated using a Cogent Diamond Hydride HILIC column (150 × 2.1 mm) at a flow rate of 0.400 µL/min. Then, 2 µL of the sample was injected with pooled, blank samples consisting of liquid chromatography–mass spectrometry (LCMS) grade water, and quality control samples were analyzed every 10 samples to measure any potential spectral drift or contamination. The solvents used were 95:5 water/ACN with 0.1% formic acid (solvent A) and 95:5 ACN/water with 0.1% formic acid (solvent B) over a 15 min elution gradient. All samples underwent standard MS1 and pooled samples underwent liquid chromatography–tandem mass spectrometry (LC-MS/MS) with a constant energy ramp of 20–50 V. Data were manually inspected to determine if any issues arose during the course of the run prior to statistical analysis.

### 2.8. Statistical Analysis

Power for this study was calculated a posteriori using plasma untargeted metabolomic data from a similar study [33]. Using this method, the sample size of *n* = 10 had high power (0.85 to 0.97) for the top 10 percentile of ranked features, which had a large average effect size (d = 0.91).

All data processing including peak picking, alignment, and metabolite annotations was conducted utilizing Water’s Progenesis QI software, v.2.3 (Nonlinear Dynamics, Newcastle, UK) and an in-house metabolite library (Mass Spectrometry Library of Standards, IORA Technologies, Ann Arbor, MI, USA). The Human Metabolome Database (HMDB) and the in-house metabolite library were used to compare acquired and theoretical fragmentation of the MS1 and MS2 data. Confident metabolite identification required a score greater than 40/100. Putative metabolite identification required a score greater than 30/100. Mass error, isotope distribution similarity, and fragmentation contribute to these scores. Additionally, features with a parts per million (ppm) error greater than 20 ppm were excluded. Multivariate statistical analysis was performed with MetaboAnalyst 6.0 (https://www.metaboanalyst.ca/home.xhtml (accessed on 15 September 2024)). Standard procedures were applied to correct for non-normal distributions while raw data were log transformed, quantile normalized, and auto-scaled (mean centered and divided by standard deviation per variable) prior to subsequent analyses. Human plasma data were analyzed via paired methods. Analyses included parallel hierarchical clustering analysis (PCHA), principal component analysis (PCA), partial least squares-discriminant analysis (PLS-DA), *t*-tests, fold change, and volcano plot analysis. ANOVA simultaneous component analysis (ASCA) was performed to evaluate treatment-/time-dependent changes [34,35]. Functional analysis used the mummichog algorithm to identify relevant metabolic pathway networks in homo sapiens (humans) [MFN] from the KEGG global metabolic and whole beef samples with mummichog version 2.0 [36]. An overview of the statistical methods employed can be found elsewhere [37].

## 3. Results

### 3.1. Participant Characteristics

Between August 2022 and March 2023, a total of 18 men and women were assessed for eligibility by completing a REDCap survey to assess eligibility criteria. Of the 18, 17 met the eligibility criteria and were invited to participate in this study. Two participants were denied entrance to the study, even when eligibility criteria were met when the number of participants per sex was previously met (five male, five female). Of the 15 participants who were entered into the study, 4 dropped out during visit 2 or 3 due to an inability to maintain intravenous catheter integrity, and one voluntarily dropped out due to conflicting commitments (Figure 1). The average age of participants was 26.6 ± 5.8 years with an average BMI of 24.44 ± 2.20 kg/m^2^. Visceral adiposity and waist circumference (cm) were significantly greater in male participants than in female participants (*p* = 0.003, *p* = 0.005). Fasting blood markers (glucose and triglyceride) did not differ between male and female participants (*p* > 0.05). Final participant (*n* = 10) characteristics for those participants successfully completing both conditions are described in Table 1. All ten participants described in Table 1 were included in the human plasma metabolomic analysis.

### 3.2. Whole Beef Steak Metabolomic Profiles

To investigate if there were differences in the global metabolomic profiles of CON versus GRA beef steaks, multivariate analysis was employed on eight steak samples. After data were filtered and corrected for non-normal distributions, 6760 metabolite features were detected in the samples. PCA revealed a significant overlap of conventional and grass-fed steaks, with PC1 and PC2 accounting for 35.6% and 15.3% of the variation in the dataset, respectively (Figure 2A). As expected, there was an overlap in the PCA. However, a supervised analysis with PLS-DA showed clear differences between CON and GRA beef. Additionally, employment of the PCHA heat map reveals perfect clustering of the two steak types, thereby suggesting that the feeding systems of cattle have a substantial effect on the metabolome. To elucidate which specific metabolites were driving the differences observed in the heat map, the PLS-DA Variance Importance in Projection (VIP) plots, heat maps, and *t*-tests were investigated (Figure 2 and Appendix A). This analysis resulted in several metabolite features that were identified (Table 2) which were best at differentiating the two feeding systems.

To further elucidate differences observed between the two steak types and to pinpoint specific biochemical pathways that differentiated CON versus GRA cattle, functional pathway enrichment analyses were performed. To do so, metabolite identifications were combined with functional analyses in MetaboAnalyst to identify pathways contributing to the differential phenotypes observed in the multivariate analysis. This approach enables the combination of predicted pathways and annotations from the KEGG database with confidently identified metabolites measured within this study. Pathway analysis revealed BCAA valine, leucine, and isoleucine degradation; ubiquinone and other terpenoid-quinone biosynthesis; nicotinate and nicotinamide metabolism; propanoate metabolism; and beta-alanine metabolism to be the most descriptive pathways. Importantly, these data, though predictive, align well with the fragmentation data obtained via mass spectrometry.

### 3.3. Human Plasma Metabolomic Profiles

To investigate the impact of beef consumption from CON and GRA cattle-feeding systems on plasma metabolomics, multivariate analysis was implemented on 180 plasma samples from postprandial timepoints. This included comparisons of metabolomic profiles between conditions at multiple timepoints throughout a 4 h postprandial time period. Of the 3803 features detected in untargeted analysis, 148 were putatively identified. All 3803 features were included in the analysis. PCA, PLS-DA, and PCHA heat map presentations of the metabolomic profile at fasting metabolomic profiles are depicted in Appendix A. PCA did not distinguish GRA and CON at fasting, while PLSDA and PCHA did. Analysis of fasting plasma features showed variation in the features present between CON and GRA conditions, although the same participants were used in both conditions per the crossover study design. Neither fasting nor postprandial PCA distinguished conditions (Appendix A). PLS-DA was able to separate the experimental groups, particularly at 1.5 h and beyond based on the 95% confidence interval ovals (Figure 3). PCHA heat maps show a comparison of metabolic profiles between cattle-feeding systems from fasting to 4 h postprandially (Appendix A). CON and GRA separate completely in PCHA heat map representation at hour 2 and separate almost completely at 2.5 and 4 h.

Based on the variation in metabolomic profiles between CON and GRA as shown in Figure 3, ASCA was implemented to identify condition/time + interaction-dependent differences/changes in feature abundance. Model validation through permutations is depicted in Appendix A. Permutation times were set to 20 based on the size of the dataset. The condition component was significant (*p* < 0.05), whereas time and interaction effects were not (*p* = 1, *p* = 0.85). Well-modeled metabolites impacted by the condition are listed in Table 3. This list is limited to features that were confidently identified, but all features were present in the analysis. These six metabolites were deemed important in regard to demonstrating a difference between conditions by the high leverage and low squared prediction error (SPE) values (on a scale from 0 to 1) observed. The scatter plots of leverage and SPE values are depicted for all three-component comparisons in Appendix A. The leverage threshold was set to 0.9 and the alpha threshold was set to 0.05. Based on these results, the condition effect impacts feature variation the most.

Functional pathway enrichment analysis was used to identify pathways contributing to metabolic variations across time. Predicted pathways that were significantly changed during the postprandial time course are shown in Figure 4. At fasting, caffeine metabolism, glycolysis, and gluconeogenesis were the most enriched pathways. Half an hour after the consumption of the steaks, tryptophan, and vitamin B12 metabolism were the most enriched. Amino acid and fatty acid metabolism are the most prevalent pathways beginning one hour after steak ingestion. At 1 h, the carnitine shuttle and saturated fatty acid beta-oxidation took place. At 1.5 h, numerous amino acid metabolism pathways (aspartate, asparagine, tryptophan, arginine, proline, BCAAs (branched-chain amino acids), lysine, and tyrosine) were present. At 2 h, the largest number of enriched pathways were found (32) that included numerous amino acid metabolism pathways like BCAAs, gluconeogenesis and glycolysis, caffeine metabolism, and glycosphingolipid biosynthesis. At 2.5 h, D4&E4-neuroprostanes formation was most prevalent, and a prostaglandin formed after the peroxidation of essential fatty acids. BCAA metabolism and gluconeogenesis/glycolysis continued to be enriched at this timepoint as well. At 3 h, lysine metabolism and BCAA degradation were the most enriched, followed by SFA beta-oxidation and fatty acid metabolism. At 3.5 h continued amino acid metabolism. Lastly, at 4 h, bile acid synthesis, squalene, and cholesterol biosynthesis, and fatty acid metabolism were most enriched, followed by amino acid metabolism. These described metabolite matches from functional pathway enrichment analysis at various timepoints are not all direct links to the identified features in the features identified from this study. The putatively identified metabolite, L-valine, was a significant hit in relation to the BCCA metabolism pathway and in greater abundance in CON. The differentiated abundance of L-valine between conditions from hours 1.5–4 is shown in Appendix A.

## 4. Discussion

To investigate metabolomic differences between GRA and CON beef, evaluation of the metabolites in samples of beef, as well as the postprandial metabolomic profiles after human consumption, were measured. We observed significant differences in metabolomic profiles of whole beef samples from GRA and CON. Metabolites that contributed to this difference in whole beef samples that were confidently identified included a greater abundance of L-threonine in GRA and a greater abundance of Proionylcarnitine in CON. We also show that GRA and CON beef samples led to distinct postprandial metabolomic profiles in consumers. Metabolic changes were observed in a number of amino acid and fatty acid-associated pathways, along with a greater abundance of L-valine in CON plasma samples. These results, measured for the first time in consumers, indicate that feeding systems cause differences in the metabolomic profiles of beef and alter the postprandial metabolism of consumers.

The hypothesis that GRA and CON cattle-feeding systems lead to different metabolomic profiles of whole beef samples was confirmed using metabolic pathway enrichment analysis. Metabolomic pathway enrichment analysis predicts functional activity by leveraging metabolic network organization, thereby bypassing metabolite identification steps [36]. By utilizing *p*-values from a *t*-test, *m*/*z* features are mapped to potential metabolites with consideration of adducts and protons [36]. These compounds are then mapped to the reference organism, and the number of features belonging to a specific pathway is reflected in the pathway enrichment analysis. The predicted pathways were corroborated metabolite identifications made as a result of the MSE data. The main pathways involved in this differentiation were valine, leucine, and isoleucine degradation; ubiquinone and other terpenoid-quinone biosynthesis; nicotinate and nicotinamide metabolism; propanoate metabolism; and beta-alanine metabolism. BCAAs (valine, leucine, and isoleucine) are essential amino acids particularly important for muscle protein synthesis and the production of Acyl-CoAs for energy production [38]. The enrichment of the BCAA degradation pathway is indicative that these amino acids play an important role in the protein makeup of CON and GRA beef. Other work has also confirmed that cattle-feeding practices do alter BCAA abundance [39,40,41,42]. Visualization of BCAA metabolism is shown in Appendix A Other metabolites that play roles in these enriched pathways and were significantly altered included threonine, tryptophan, histamine derivatives, leucyl phenylalanine, and serine. Threonine, serine, histamine derivative, and leucyl phenylalanine were in significantly higher abundance in GRA compared to CON whole beef steaks. These metabolic pathways imply changes in protein metabolism reflective of the cattle-feeding systems. This concurs with findings that feeding systems impact protein metabolism [11,43].

In addition to protein metabolism, differences between GRA and CON whole beef_steak samples in pathways relating to differences in energy metabolism and buffering were identified. Several species of carnitines and fatty acids were found to be significantly impacted by the different cattle-feeding systems (Table 2). Carnitines play a vital role in the transportation of long-chain fatty acids into the mitochondria where they are then oxidized to produce ATP [44]. Visualization of the role carnitines play in fatty acid transportation is provided in Appendix A. This observation has been noted in another study examining cattle-feeding systems and may suggest that grazing systems affect fat content and/or deposition; however, further evaluation of carnitines and long-chain fatty acids is required to appreciate this relationship between feeding systems and metabolism in cattle [45]. Carnitine metabolites are also biomarkers of red meat consumption and increased cardiovascular disease (CVD) risk [46]. Proionylcarnitine was found to be in higher abundance in CON whole beef samples. This metabolite would need to be specifically measured in plasma samples to associate cattle-feeding systems with increased CVD risk. Our finding that beta-alanine metabolism was a pathway differentiating GRA and CON whole beef steak samples is consistent with other findings in beef steak metabolomic analyses [45]. Beta-alanine functions by combining with histidine to generate carnosine, a scavenger of reactive oxygen species, pH buffering capabilities, and metal-ion chelation, and is considered a building block of protein in the body [47,48]. Additionally, it can increase the acetylation of histones, thus regulating gene expression at the epigenetic level [49]. Therefore, the presence of this pathway in the differential analysis of CON versus GRA steaks may be reflective of differences in the need for protein and energy. This may be further supported by the presence of nicotinamide metabolism in this study. Nicotinamide (NAM) is the precursor for the coenzyme nicotinamide adenine dinucleotide (NAD). NAD is an active participant in energy metabolism as it plays roles in oxidation–reduction reactions [50]. Although intriguing, more research is required to formulate robust conclusions.

Functional pathway enrichment analysis revealed that throughout the postprandial time course, pathway enrichment fluctuated, reflective of digestive timing and differentiated metabolites between conditions. This is the first time that metabolic profiles and related metabolic pathways have been measured in response to the consumption of beef. An hour after the consumption of steaks, various amino acid and fatty acid metabolism pathways were enriched. The timing of this coincides with another study observing protein and lipid digestion at one and two hours post-consumption [12]. BCAA metabolism was an enriched pathway for the majority of the postprandial time period (hours 1.5–4) (Figure 3). Saturated fatty acid (SFA) beta-oxidation was also an enriched pathway at multiple timepoints (0, 1, 3, and 4 h). It is not consistently reported whether GRA or CON beef has a greater content of SFA [4,5]. SFAs have been associated with increased systemic inflammation and heart disease but are also physiologically important to cellular functions such as N-terminal myristoylation and the transcription of lipogenic genes [51,52,53]. SFAs are most commonly consumed from meats and milk, and the enrichment of this pathway suggests that cattle-feeding systems are impacting the prevalence of SFAs being metabolized [54]. Caffeine metabolism was also an enriched pathway at hours 0, 2, 2.5, and 4. Participants were offered caffeinated black tea during the study visits to mitigate caffeine withdrawal symptoms. Knowing that caffeine was given to participants validates the sensitivity and strength of the analysis carried out.

Analysis of human plasma samples also showed that postprandial metabolomic profiles of GRA and CON significantly differed from each other. Between conditions, Calamendiol was found to be one of the differentiating metabolites (Table 3). Calamendiol is a sesquiterpenoid compound and a known metabolite of the plant *Acorus calamus* L. [55]. This compound has been found to have anti-inflammatory potential, indicating its potential as a health-promoting biomarker from cattle exposed to this plant [56]. It was determined that condition differences contributed most to significant differences in metabolomic profiles compared to time or the interaction of condition and time. This is indicative that cattle-feeding systems do have an impact on the human metabolome, as it did in the whole beef samples. L-valine, a BCAA, was found to be in greater abundance in CON (Appendix A). L-valine is a metabolite of valine that can be produced by fermentation of *Escherichia coli* and *Corynebacterium glutamicum* [57,58]. Increased plasma concentrations in humans of L-valine have been associated with increased oxidative stress [59]. Excessive oxidative stress can be harmful to a person’s health and plays a role in the development of cardiovascular and neurological diseases due to the resulting cellular damage [60]. BCAAs have also been associated with increased CVD risk due to the role they can play in the development of type 2 diabetes [46]. L-valine was also found to be a differentiating metabolite between conditions according to the ASCA (Table 3). Along with BCAA metabolism, alanine, aspartate, asparagine, lysine, tryptophan, arginine, and tyrosine metabolism were among the enriched pathways. This shows that metabolites with metabolic are impacted by CON and GRA beef.

This study demonstrates new advancements in the measurement of metabolic responses to consuming beef; however, human participant research includes some limitations. The participant population used in this clinical trial included both men and women of a healthy BMI with no preexisting conditions. This population was chosen to establish a baseline for postprandial response to consuming these specific types of beef, as this is one of the first studies to investigate this response. Male and female participants were not expected to have different responses to consuming beef, based on previous research, which was confirmed by the results as well [61]. The fasting metabolomic profiles, as shown in Appendix A, did show the separation of metabolites by treatment, contrary to the expected mixed variation between subjects rather than within subjects. Efforts to control differences within subjects between study visits included instructions to fast for 12 h prior to the time of blood draws and to consume the same meals in the 24 h prior to each study visit (visits 2 and 3). Despite these instructions, caffeine metabolism was one of the most enriched pathways present at fasting per the functional analysis. This could also be a carryover from the previous day, particularly if caffeine was consumed later in the day, as the half-life of caffeine is 3–10 h [62]. This does show the strength and sensitivity of the functional analysis methods and detection of features related to caffeine metabolism with LC/MS since we can link this finding back to the enriched caffeine metabolism at hours 2 and 2.5 resulting from the ingestion of black tea around hour 0.5. With human nutrition research, errors related to participants compliance are always possible and our results may indicate a deviance by participants from the instructed protocol. The sample size of the study, 10 participants, also may contribute to the unexpected variations within subjects during the 2-week testing period. Each subject accounts for 10% of the total data, as it was a crossover design and data were collected from each subject on two occasions. Daily variations in the human metabolome are expected, so the sample size may amplify these natural variations despite the efforts to control for these [63]. Interpretation of the human plasma metabolic profiles can be generalized to individuals who represent the study population of healthy men and women with no preexisting conditions and a healthy BMI. Natural limitations of untargeted metabolomic analysis exist; for instance, the large sample size and thousands of observed features hinder the ability to identify the majority of features [64]. Further targeted analysis needs to be conducted to elucidate specific metabolites that differentiate metabolic profiles seen after consuming GRA and CON beef that were shown in these data. We did not assess total protein, albumin, urea, or creatine in this study. We acknowledge that future research in this area would benefit from the addition of these measures to gain insight into the link between protein, glucose, and fat metabolism [65]. In both PCA and PLS-DA visualization, two conditionings independent of CON and GRA conditions are apparent at all timepoints. This was not due to the separation of male and female participants or baseline participant anthropometrics and diet. This could be the result of batch effects; however, this does not seem to confound the results. Manual inspection and alignment of the raw MS data were performed in an attempt to minimize this possibility. Since the separation lies within the 95% confidence intervals of each condition, we are confident that it is not majorly contributing to the separation between CON and GRA that was observed. A strength of these findings is that participants were fed an unseasoned steak, not combined with a multi-component meal. This supports our findings that the beef itself is what caused the separation in metabolomic profiles, and not confounding biomarkers from other foods.

## 5. Conclusions

In conclusion, these results provide evidence that the cattle-feeding systems, GRA and CON, impact the metabolomic profiles of both whole beef and human postprandial plasma samples. We observed differences in metabolite abundance between CON and GRA beef that are associated with health impacts such as oxidative stress and inflammation. After consuming the beef, we observed metabolic pathway enrichment changes through the postprandial time course, reflecting changes induced by beef consumption on fatty acid and amino acid metabolism. These novel findings help to clarify the impact that consuming beef has on metabolites and metabolic pathways, although the analysis does not distinguish one condition as more favorable than the other. These findings can be used to inform future studies with more participants and/or having metabolic dysregulation regarding long-term consumption of beef from various cattle-feeding systems to further understand the downstream impacts the agricultural practices have on consumers. Further work will be useful to also elucidate more specific differences about which type of beef would be the preferred choice for consumers.

## Figures and Tables

**Figure 1 metabolites-14-00533-f001:**
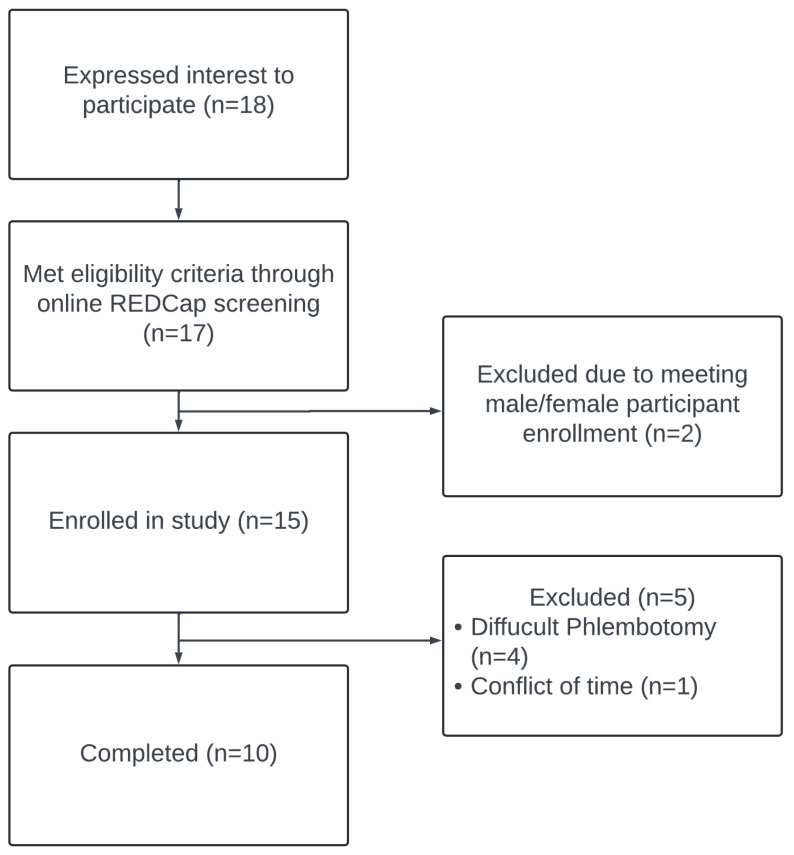
CONSORT flow chart.

**Figure 2 metabolites-14-00533-f002:**
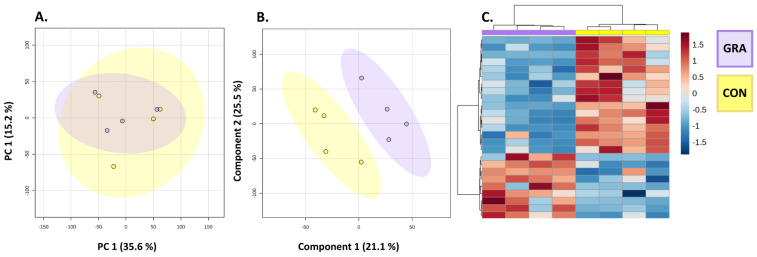
Multivariate statistics of CON versus GRA steaks. (**A**) PCA scores plot of CON (gold) and GRA (violet) beef steaks. PC1 describes 35.6% of total variation. PC2 represents 15.2% of total variation. Ovals represent 95% confidence intervals. (**B**) PLS-DA scores plot of CON versus GRA beef steaks. PC1 described 21.1% of total variation. PC2 described 25.5% of total variation. Ovals represent 95% confidence intervals. (**C**) PCHA heat map of the top 25 features, filtered by lowest *p*-value. Samples are clustered in columns and features are clustered in rows. The intersection represents the feature abundance of the sample, relative to the average. Red indicates high abundance and blue represents low abundance.

**Figure 3 metabolites-14-00533-f003:**
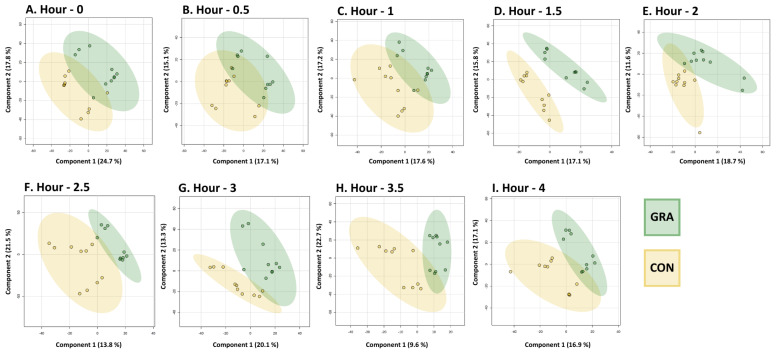
PLS-DA showing postprandial metabolomic profiles in human plasma (*n* = 10) between CON and GRA at (**A**) fasting, (**B**) hour 0.5, (**C**) hour 1, (**D**) hour 1.5, (**E**) hour 2, (**F**) hour 2.5, (**G**) hour 3, (**H**) hour 3.5, and (**I**) hour 4. All features included in analysis. Ovals represent 95% confidence intervals. All features were included to generate these plots. The greatest separation in postprandial metabolic profiles between GRA and CON occurred at hour 1.5, followed by moderate separation in hours 2–4.

**Figure 4 metabolites-14-00533-f004:**
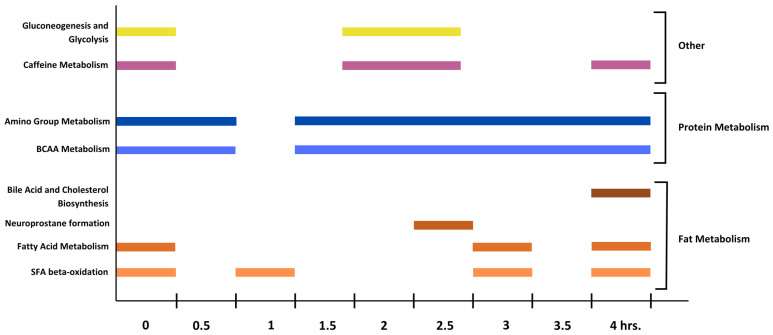
Time course of enriched pathways from functional pathway enrichment analysis. All features included in analysis. Pathways altered based on the untargeted metabolomics data (0–4 h) are shown. Fat and protein metabolism dominated metabolic pathways that were enriched after the consumption of CON and GRA beef. Horizontal bars coincide with the timepoint(s) when the pathway was enriched. Color shading (orange, blue, green, purple, yellow) relates to the type of metabolism pathway.

**Table 1 metabolites-14-00533-t001:** Final participant (*n* = 10) baseline characteristics. Fasting glucose and triglyceride values are presented and represent the average fasting value from both treatment visits. Values are presented as mean ± standard deviation (SD).

Participants	Age (Years)	BMI (kg/m^2^)	WC (cm)	VAT (L)	Glucose (mmol/L)	Triglyceride (mmol/L)
All (*n* = 10)	26.6 ± 5.8	24.44 ± 2.20	82.33 ± 8.32	0.78 ± 0.73	5.47 ± 0.81	0.98 ± 0.45
Female (*n* = 5)	24.8 ± 5.2	23.99 ± 2.38	76.77 ± 7.34	0.34 ± 0.09	5.52 ± 1.14	0.80 ± 0.22
Male (*n* = 5)	28.4 ± 6.3	24.89 ± 2.16	87.88 ± 4.97 *	1.22 ± 0.81 *	5.41 ± 0.32	1.15 ± 0.56

* Indicates a *p* value < 0.05 between male and female subjects. BMI; body mass index, WC; waist circumference, VAT; visceral adipose tissue.

**Table 2 metabolites-14-00533-t002:** Differentiating metabolites from whole beef steak analyses (*n* = 4). All features included in analysis. These metabolites show that differences between CON and GRA beef are present, specifically in amino acid and carnitine species.

Metabolite	Score ^1^	Raw *p*-Value	CON vs. GRA ^2^
9-Hexadecenoylcholine	39.2	3.2919 × 10^−4^	CON > GRA
L-threonine	28.6	0.0035	CON < GRA
Keratan sulfate II	38.4	0.0051	CON > GRA
3-hydroxy-5,8-tetradecadienoylcarnitine	34.9	0.0236	CON > GRA
Proionylcarnitine	37.1	0.0021	CON > GRA
Aspartyl-Serine	34.3	0.0251	CON < GRA
Glycyl–Phenylalanine	37.1	0.0488	CON < GRA
L-Tryptophan	38.7	0.0419	CON > GRA
Histamine-betaxanthin	38.6	0.0366	CON < GRA
Leucylphenylalanine	40.1	0.0416	CON < GRA
N-palmitoyl serine	38.7	0.0276	CON > GRA
LysoPC(18:1/0:0)	37.9	0.0474	CON < GRA
DIBOA trihexose	37.9	0.0330	CON > GRA

^1^ Identification score; ^2^ indicates which condition the metabolite was in higher abundance.

**Table 3 metabolites-14-00533-t003:** Differentiating metabolites from human plasma ASCA of the condition component (*n* = 10). All features included in analysis. These identified metabolites show that consumption of GRA and CON beef elicit differences in the postprandial metabolome.

Metabolite	Score ^1^	Leverage	SPE ^2^
L-valine	38.7	0.0017	2.7733 × 10^−31^
Calamendiol	46.5	0.0020	2.7733 × 10^−31^
5-Aminopentanal	39.4	0.0020	2.7733 × 10^−31^
3-Amino-4,7-dihydroxy-8-methylcoumarin	39.1	0.0015	6.9333 × 10^−32^
3-beta-D-glucopyranuronosyloxy-5-methylisoxazole	39.1	0.0013	0
PE(16:1(9Z)/20:5(5Z,8Z,11Z,14Z,17Z)	37.8	0.0014	0

^1^ Identification score; ^2^ squared prediction error (SPE).

## Data Availability

The metabolomic datasets presented in this study can be found in online repositories. Metabolomic data including raw files may be found at https://www.metabolomicsworkbench.org (accessed on 27 August 2024). Other data are not publicly available due to participant confidentiality.

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
