# Peer review of "Comparative Impact of Organic Grass-Fed and Conventional Cattle-Feeding Systems on Beef and Human Postprandial Metabolomics—A Randomized Clinical Trial"

_metabolites, 2024, doi:10.3390/metabo14100533_

Round 1
Reviewer 1 Report
Comments and Suggestions for Authors
-Nice work, from well know lab.s, Univ.s and researchers….., and I appreciate a lot for focusing on this important experimentally and clinically applicable subject. Knowledge on the metabolomic profiles of beef and knowing and improving postprandial metabolomic profiles in food animals and consumers are very valuable. However, it needs some revision and improvement
-The title is very broad and somehow should be more relevant to what has been done here. Here you specifically used two conditions, GRA beef vs. CON beef! As such, I would use …. “Comparative Impact of Organic Grass-Fed and Conventional Cattle Feeding systems on Beef and Human Postprandial Metabolomics-A Randomized Clinical Trial” as title in your revised version.
-Line 162 and throughout the text for centrifugation please use ×g for RPM.
-Line 188, ….were spun at… spun here is not appropriate word scientifically (for 10 min. of centrifugation). Please elaborate here and throughout the text (if any).
-Throughout the text, please apply word “use” for “utilize”.
-Some issues related to the abbreviations…, indeed, it not necessary to redefine more than once throughout the text (e.g., PLS-DA, PCHA, ASCA etc.); please check throughout the text and clearly elaborate. Or for CON and GRA when you defined first then latter in the text no need to rewrite the complete name(s) please check and correct them throughout the text.
-In the captions of the tables 2 and 3 please mention what is your main message and also the n=?
-The same for the captions of the figs 2 and 3 (please mention what is your main message and also the n=?).
-Lines 363-365, it is not discussion; please remove from here.
-The discussion is broad and somewhat beyond the focus. Please be more specified throughout the text and avoid some speculative notion. I would also draw a graphical figure about the work (specialized scheme to show some mechanistic points for simplicity and clarity) and particularity for visual effects.
Very good luck
Reviewer 2 Report
Comments and Suggestions for Authors
Dear authors,
the manuscript “Impact of Cattle Feeding Systems on Beef and Human Postprandial Metabolomics – A Randomized Clinical Trial” (metabolites-3211450), by Meghan Spears and the coauthors, is devoted to an interesting and important problem of metabolomics and clinical trials. The aims of this study were the following: to measure whole beef metabolomics and postprandial metabolomic response of consumers between “GRA and CON beef”, to elucidate potential health implications, to use a “double-blind, crossover design with healthy male and female subjects”, etc. The manuscript analyzed the literature works in detail and at high level of discussion. It is positive that the authors used the combined method of liquid chromatography–mass spectrometry (LC-MS) for the “untargeted metabolomic analysis of whole beef and human serum samples” (from 10 healthy males and females), as well as the “multivariate and pathway enrichment analysis in MetaboAnalyst” in order to investigate metabolite and biochemical pathways that distinguished organic grass-fed (GRA) and conventional (CON) cattle feeding systems. These cattle feeding systems impacted both postprandial and whole beef steak metabolomic profiles. The investigated metabolites (including carnitine species, fatty acids, some L-amino acids and calamendiol) have been associated with oxidative stress, inflammation, and cardiovascular health. Functional pathway enrichment analysis revealed numerous amino acid degradation pathways, especially branched chain amino acids (BCAA), and fatty acid degradation that changed throughout the postprandial time course. These findings suggest that CON and GRA cattle feeding systems differentially impact whole beef metabolomics, as well as consumer postprandial metabolic responses and the associated health implications. I do not doubt the technical quality of the work and feel that there is a sufficient impact on a broader readership to justify publication in the "Metabolites". All these results are in the frame of the journal scopes; the subject matter is treated in depth. Thus, the present manuscript is important and actual.
There are some comments:
1. The only limited literature works are analyzed in the part 1 “Introduction” in details (28 from the 61 total). It will be reasonable to increase the number of the cited literature works (in some cases) and the level of their discussion in the part 1 “Introduction”.
2. It is important to add in the part 3. “Results 3.1. Participant characteristics” (page 6, lines 255-259) in the Table 1 (“Final participant (n=10) baseline characteristics….”) the info about the average values of the following biochemical indicators of human blood serum samples: total protein (TPOT); albumins; urea; creatinine, etc. These biochemical indicators are very important for characterizing the protein metabolism in addition to the glucose and triglyceride values that are presented in the Table 1 and characterizing the glucose and lipid metabolism.
3. It is important to add in the part 3. “Results 3.3. Human Serum Metabolomic Profiles” (page 8, lines 332-334) in the Table 3.(“Differentiating metabolites from human serum ANOVA simultaneous component analysis (ASCA) of the condition component…”) some data about another amino acids (such as aspartate, asparagine, tryptophan, arginine, proline, branched chain amino acids, lysine, and tyrosine) that are discussed at lines 342-343.
4. It will be useful to include some general metabolic schemes concerning these substances in the part 4. “Discussion”.
Accept after minor revision.

Minor editing of English language required.
